# A classification model of homelessness using integrated administrative data: Implications for targeting interventions to improve the housing status, health and well-being of a highly vulnerable population

**Thomas Byrne** [1]*, **Travis Baggett** [2,3], **Thomas Land** [4], **Dana Bernson** [5], **Maria-Elena Hood** [5], **Cheryl Kennedy-Perez** [5], **Rodrigo Monterrey** [5], **David Smelson** [4], **Marc Dones** [6], **Monica Bharel** [5]

1 Boston University School of Social Work, Boston, Massachusetts, United States of America, 2 Boston Health Care for the Homeless Program, Boston, Massachusetts, United States of America, 3 Division of General Internal Medicine, Massachusetts General Hospital, Boston, Massachusetts, United States of America, 4 University of Massachusetts Medical School, Worcester, Massachusetts, United States of America, 5 Massachusetts Department of Public Health, Boston, Massachusetts, United States of America, 6 National Innovation Service, United States of America

* tbyrne@bu.edu

**Data Availability Statement:** Data cannot be publicly shared due to legal restrictions that

## Abstract

Homelessness is poorly captured in most administrative data sets making it difficult to understand how, when, and where this population can be better served. This study sought to develop and validate a classification model of homelessness. Our sample included 5,050,639 individuals aged 11 years and older who were included in a linked dataset of administrative records from multiple state-maintained databases in Massachusetts for the period from 2011–2015. We used logistic regression to develop a classification model with 94 predictors and subsequently tested its performance. The model had high specificity (95.4%), moderate sensitivity (77.8%) for predicting known cases of homelessness, and excellent classification properties (area under the receiver operating curve 0.94; balanced accuracy 86.4%). To demonstrate the potential opportunity that exists for using such a modeling approach to target interventions to mitigate the risk of an adverse health outcome, we also estimated the association between model predicted homeless status and fatal opioid overdoses, finding that model predicted homeless status was associated with a nearly 23-fold increase in the risk of fatal opioid overdose. This study provides a novel approach for identifying homelessness using integrated administrative data. The strong performance of our model underscores the potential value of linking data from multiple service systems to improve the identification of housing instability and to assist government in developing programs that seek to improve health and other outcomes for homeless individuals.

prohibit the sharing of both the data sources used in the construction of the study's analytic data set and the analytic data set itself. More specifically, study authors had access to these data by responding to a Notice of Opportunity issued by the Commonwealth of Massachusetts that enabled interested parties to propose analysis of these data for a set period of time. The deadline for responding to the Notice of Opportunity was April 30, 2017. Information about that Notice of Opportunity is available at the following link: https://www.commbuys.com/bso/external/bidDetail.sdo?docId=BD-17-1031-HISRE-HIS01-11089&external=true&parentUrl=bid The Commonwealth subsequently issued a second Notice of Opportunity to conduct analysis with these data, and applications in response to that Notice of Opportunity closed on March 23, 2018. Additional information about that Notice of Opportunity is available here: https://www.commbuys.com/bso/external/bidDetail.sdo?bidId=BD-18-1031-OFFIC-ODMOA-24680&parentUrl=activeBids At the time of the submission of this manuscript, there Commonwealth of Massachusetts was not an open Notice of Opportunity to which interested researchers could apply to access these data. Additional information about the data used in this study and relevant legal restrictions and data access are available at the following link: https://www.mass.gov/public-health-data-warehouse-phd As per that website, the contact person for the data are Abigail Averbach (Abigail.Averbach@MassMail.State.MA.US) and Brigido Ramirez Espinosa (Brigido.Ramirez1@State.MA.US) at the Massachusetts Department of Public Health Office of Population Health.

**Funding:** One author (Marc Dones) was employed by a commercial entity, the Center for Social Innovation, at the time work on the manuscript was completed. The funder provided support in the form of salaries for authors [insert relevant initials], but did not have any additional role in the study design, data collection and analysis, decision to publish, or preparation of the manuscript. The specific roles of this author are articulated in the 'author contributions' section.

**Competing interests:** I have read the journal's policy and the authors of this manuscript have the following competing interests: Travis Baggett receives royalties from UpToDate for authorship of a topic review on health care for homeless people. This does not alter our adherence to PLOS ONE policies on sharing data and materials. Marc Dones was employed by a commercial entity, the Center for Social Innovation, at the time work on this

## Introduction

Homelessness is associated with a wide range of adverse social, economic and health outcomes [1–3]. Persons experiencing homelessness often interact with multiple publicly-funded systems of care including the emergency shelter, health care, mental health, substance use disorder treatment, and criminal justice systems [4–7], thus providing numerous points to address their housing, health care, and other social needs.

However, the capacity of publicly-funded systems to intervene is hampered by unavailable or incomplete data regarding persons who are experiencing homelessness or housing instability. Accurate risk identification even without a perfect record of periods of homelessness would still enhance the potential for the more effective targeting of a variety of housing, health care, and social service interventions. Unfortunately, many service systems do not capture information about housing status in a reliable manner, despite the potential importance of such information for tailoring service delivery to those experiencing housing instability. Recognition of this shortcoming has led to increased interest in developing predictive models to identify persons experiencing homelessness using available data in administrative records. Much of this work has been conducted in health care systems where studies have used indicators obtained from medical records, including diagnosis codes [8], address information [9, 10], and free text notes [11–13], to develop models identifying persons experiencing homelessness. Yet, these studies are limited by their exclusive reliance on data obtained from medical records and thus are based on a limited set of predictor variables and apply to non-representative samples of individuals.

Only one study [14] has used administrative records from multiple service systems to identify predictors of homelessness. However, that study focused on a specific population (young adults exiting foster care) and sought to identify salient predictors of homelessness rather than evaluate the overall performance of a predictive model that might be used in an applied context. Two related studies [15, 16] have used linked administrative data from multiple service systems to develop predictive models of the risk of long-term homelessness among two specific sub-populations (low-wage workers who recently experienced a job loss and young adults receiving public assistance) and of extremely high cost use of public services among individuals currently experiencing homelessness population, respectively. However, to our knowledge, no study to date has used multiple sources of administrative data from public service systems to develop and evaluate the performance of a similar model of homelessness in a broader population.

Therefore, the current study capitalizes on the availability of a unique and rich data source that integrates administrative records from a wide array of service systems in Massachusetts to develop and test a classification model of homelessness. Because of widely recognized challenges in accurately identifying people experiencing homelessness [17] and evidence that administrative data sources capturing housing status do not fully concord with self-reported housing status [18] this study makes the assumption that our data on homelessness is incomplete and that an examination of patterns of relationships between known cases of homelessness and other data can be applied to individuals who have not been recorded as having experienced a period of homelessness. As a second and exploratory aim intended to illustrate the potential opportunity for health improvements that that exists from using a similar modeling approach in an applied context as a mechanism to target interventions to mitigate serious health outcomes, we use the results of this classification model to assess the relationship between homelessness and risk of fatal opioid-related overdoses, a particularly adverse outcome previously linked to homelessness [19] that is especially important to evaluate in light of increasing rates of opioid-related overdose deaths nationwide [20].

manuscript was completed. This does not alter our
adherence to PLOS ONE policies on sharing data
and materials. No other authors have any
competing interests to disclose.

# Methods

## Data and sample

Data for the present study come from the Massachusetts Chapter 55 of the Acts of 2015
("Chapter 55") integrated dataset. Enacted in August 2015, Chapter 55 authorized the linkage
and mandated the analysis of several Massachusetts government databases to better under-
stand the opioid epidemic and guide policy development. Chapter 55 allowed the Massachu-
setts Department of Public Health (DPH) to link individual-level records from 16 state-based
administrative data sources. Persons aged 11 years or older who had health insurance between
2011 and 2015 as reported in the Massachusetts All-Payer Claims Database (APCD), which
aggregates health care claims from all public and private payers, comprised the universe of
individuals included in the Chapter 55 data warehouse. APCD data for these individuals, who
represent more than 98% of Massachusetts residents, were linked with other datasets using a
multistage deterministic linkage algorithm; full details on this linkage algorithm and about the
16 data sources included in the Chapter 55 data warehouse are available elsewhere [21]. The
present study used data from 15 of the 16 sources that contributed to the Chapter 55 dataset
(see Table 1). Since many of the 15 data sets included the same demographics, a "master demo-
graphic" dataset was created from the best available demographic information from across all
Chapter 55 datasets.

The Chapter 55 dataset included records for a total of 14,245,349 individuals, based on
APCD data. This exceeded the actual number of Massachusetts residents who met criteria for
inclusion in the Chapter 55 dataset, suggesting that a number of records in the data reflected
either non-Massachusetts residents or unresolved duplicate records. To ensure that our sample
included only unique Massachusetts residents, we conservatively limited the cohort for the
present study from the 14,245,349 unique individuals in the APCD data to the 5,050,639
unique individuals who had a at least one record in the APCD and one other Chapter 55 data-
set. See S1 Fig for a schematic diagram showing sample selection process.

## Measures

**Measures of homelessness.**  Developing a classification model of homelessness required
that we identify known cases of homelessness in the Chapter 55 datasets. Based on the consen-
sus of a working group of experts in homelessness, we used the following criteria to identify
these known cases: 1) a claim in the APCD or record in the Acute Care Hospital Case Mix
(Case Mix) data with an accompanying ICD-9 V.60 or ICD-10 Z590 code indicating homeless-
ness; 2) a record in the Department of Mental Health (DMH) dataset in which individuals
were ever identified as experiencing a loss of housing based on a measure of housing status
captured on a monthly basis for all DMH clients; 3) an ambulance record in the Massachusetts
Ambulance Trip Record Information System (MATRIS) data in which the word "homeless" or
"shelter" appeared in the narrative report; or 4) a prescription record in the Prescription Moni-
toring Program (PMP) in which the patient's address matched that of an emergency shelter.
Individuals meeting any of these criteria at any point during the 5-year observation period
were classified as experiencing homelessness.

**Independent variables.**  We selected 94 possible independent variables from across all 16
Chapter 55 datasets based on prior research identifying correlates of homelessness [22–25].
These predictors were classified into several groups, including socio-demographic predictors
(e.g. age, gender, race, Medicaid receipt [a proxy for socioeconomic status]); drug/alcohol use
predictors (e.g. presence of drug/alcohol diagnoses, use of substance use disorder treatment
services); mental health predictors (e.g. presence of mental health diagnoses, use of mental

**Table 1. Summary of chapter 55 datasets and variables included in classification model.**

| Data Source | Description | Sample Variables |
|---|---|---|
| Chapter 55 Master Demographic Dataset | Dataset aggregating and reconciling demographic information from all Chapter 55 datasets | • Age<br>• Sex<br>• Race/ethnicity |
| Massachusetts All Payer Claims Database (APCD) | Health, pharmacy, and dental insurance claims for the ~80 private health care payers, public health care payers and publicly supported managed care organizations and senior care organizations across Massachusetts. | • Indicator of MassHealth (Massachusetts Medicaid program) membership<br>• Separate indicators of any claims with diagnoses for:<br>• Psychoses<br>• Schizophrenia<br>• Substance use disorders<br>• Opioid use disorder<br>• Alcohol use disorder |
| Massachusetts Cancer Registry | Population-based registry tracking incidence of cancer. | • None included, used only to restrict sample |
| Acute Care Hospital Case Mix | Records for all inpatient, emergency department, and outpatient observations discharges from acute care hospitals in the state | • Indicator for any use of emergency department services<br>• Separate indicators for any inpatient, emergency department and outpatient observation discharge with claims with diagnosis codes for:<br>• Skin/soft tissue infection<br>• Anxiety disorder<br>• Bipolar disorder<br>• Medication induced mental health disorder<br>• Injection drug use<br>• Obsessive compulsive disorder |
| Massachusetts Department of Correction (DOC) | Records for individuals incarcerated in Massachusetts prisons | • Indicator for any history of incarceration in DOC facility |
| Massachusetts Department of Housing and Community Development (DHCD) Emergency Assistance Program | Records of heads of homeless families who received services from the Emergency Assistance program. | • None included, used to restrict sample |
| Massachusetts Department of Mental Health (DMH) | Records for individuals receiving services from DMH, the Massachusetts State Mental Health Authority. | • Indicator of psychiatric hospitalization<br>• Indicator of incarceration, as recorded by DMH |
| Massachusetts Department of Veteran Services (DVS) | Records for individuals receiving medical, housing, or other benefits from DVS | • Indicator of receipt of medical benefits from DVS |
| Massachusetts Department of Public Health, Bureau of Substance Addiction Services (BSAS) | Substance use disorder (SUD) treatment episode data from BSAS-funded SUD treatment providers. | • Separate indicators for BSAS-funded services including:<br>• Detox<br>• Case management<br>• Post-Detox treatment<br>• Outpatient treatment |
| Massachusetts Ambulance Trip Record Information (MATRIS) | Emergency medical service (EMS) incident data from licensed ambulance services. | • Indicator for any ambulance trip |
| Massachusetts Department of Public Health, Prescription Monitoring Program (PMP) | Records for prescriptions for schedule II through V medications filled by all Massachusetts community, hospital outpatient, and clinic pharmacies as well from out-of-state mail order pharmacies delivering to Massachusetts. | • Indicator for Veteran status as recorded in PDMP |
| Massachusetts Office of the Chief Medical Examiner (OCME) Intake forms | Cause of death | • Opioid related deaths |
| Massachusetts Office of the Chief Medical Examiner (OCME) Toxicology Reports | Toxicology Reports | • Opioid related deaths |
| Massachusetts State Police | Circumstances of Death Reports | • Opioid related deaths |
| Massachusetts Registry of Vital Records and Statistics (RVRS) Death Records | Official death certificates | • Opioid related deaths |
| Massachusetts Registry of Vital Records and Statistics (RVRS) Birth Records | Official birth certificates | • Mother's occupation code |

health services); physical health predictors (e.g. skin disorders); other service use predictors (e.g. history of incarceration in state prison, use of emergency department services). Table 1

provides examples of these predictors from each of the Chapter 55 datasets (the full set of predictors are provided in S1 Table).

**Fatal opioid overdoses.**   Fatal opioid-related overdoses were identified from death records from the Massachusetts Registry of Vital Records and Statistics (RVRS). Deaths were classified by using the International Classification of Disease (ICD-10) codes for mortality or using a literal search of written cause of death from the medical examiner's office for records that did not yet have an ICD-10 code assigned. The following codes were selected from the underlying cause of death field to identify poisonings/overdoses: X40-X49, X60-X69, X85-X90, Y10-Y19, and Y35.2. All multiple cause of death fields were then used to identify an opioid-related death, which included any of the following ICD-10 codes: T40.0, T40.1, T40.2, T40.3, T40.4, and T40.6

## Analysis

Our primary aim and analytic plan centered on the development and testing of a classification model of homelessness. The terms "predictive model" and "classification model" are frequently used interchangeably to describe models that attempt to predict a categorical outcome conditional on a set of predictors with the goal of maximizing the performance of such models. Although these terms are often used interchangeably, we chose to use the term "classification model" in the current paper to reflect the fact that our data are cross-sectional, and thus we cannot determine the temporal ordering of our outcome variable (homelessness) relative to our set of independent variables. Thus, our analysis is not "predicting" a future outcome on a set of antecedent predictors, and we use the term "classification" model to avoid confusion about the scope of our analysis.

To develop and test our classification model, we split the study sample into a development sample to be used in building the classification model of homelessness and a validation sample to be used to evaluate model performance. Given the proportionally small number of cases in our dataset identified as homeless and to ensure that the development and validation samples included equal proportions of individuals identified as homeless, we used a stratified random sampling approach to divide the sample into a development and validation sample. Specifically, we identified two strata based on whether individuals were identified as homeless based on the criteria outlined above. We then randomly assigned 75% of the cases within each stratum to the development sample and the remaining 25% of cases within each stratum to the validation sample. This resulted in a development sample that comprised 75% (n = 3,787,980) of cases in the full sample while the remaining 25% of cases from the full sample (n = 1,262,659) formed the validation sample.

We used multivariable binary logistic regression as the classification method in developing our classification model of homelessness. We initially estimated a model that included all individuals in the development sample. However, the small proportion of individuals in our cohort who met the criteria for homelessness (0.82%) resulted in models that had near perfect specificity but extremely poor sensitivity. We therefore used a technique called downsampling to balance outcome class membership in the development sample [26]. In the present context, downsampling worked by retaining all persons identified as homeless in the development sample and then randomly selecting an equal size number of persons not identified as homeless for inclusion, while excluding all other cases. We then used this balanced development sample in the model development phase.

We applied parameter estimates from the logistic regression model estimated using the development sample to derive predicted probabilities of homelessness for all individuals in the validation sample. We evaluated model performance using area under the receiver operating

curve (AUC), sensitivity (i.e. true positive rate), specificity (i.e. true negative rate), and balanced accuracy, which is the average proportion of correctly classified cases in each outcome category and is a better metric of overall model accuracy when there is severe imbalance between outcome classes [27]. We also calculated positive predictive value, which measures true positives as a proportion of all model predicted positive cases and negative predictive value, which models true negatives as a proportion of all model predicted negative cases.

To address the study's second aim, we estimated fatal opioid-related overdose rates per 100,000 persons for both homeless and non-homeless individuals in the validation sample using model predicted probabilities to classify persons as homeless or not homeless. In the principal analysis, we classified individuals with predicted probabilities of ≥0.5 as homeless and individuals with predicted probabilities of <0.5 as non-homeless. In sensitivity analyses, we used two alternative approaches for assigning homelessness status based on model-predicted probabilities. In the first approach, we assigned all persons in the validation cohort with a known case of homelessness (regardless of their model predicted probability) a risk score of 1, and we used the model-predicted probabilities as the risk score for all other members of the study cohort. In the second approach, we assigned all persons with a known case of homelessness (based on criteria described above) a risk score of 1 and all persons with no observed homeless indicator and a model predicted probability <0.5 a risk score of 0, with all remaining individuals assigned a risk score equivalent to their predicted probability.

We then used these risk scores to calculate weighted estimates of the number of homeless and non-homeless persons in the validation sample in addition to the number of fatal opioid overdoses experienced by each group. Specifically, we calculated the weighted estimate of the number of homeless persons as the sum of the homeless risk scores for all those with scores ≥0.5 and the weighted estimated number of non-homeless persons as the sum of the inverse of the homeless risk scores for all those with risk scores <0.5. We calculated the weighted estimate of the number of overdoses in the homeless group using a two-step process. First, we multiplied the homeless risk scores for all those with scores greater than 0.5 by 1 if they experienced a fatal overdose or 0 if they did not. We then summed the resulting products to estimate the number of fatal overdose deaths in the homeless groups. To estimate the number of fatal overdose deaths in the non-homeless group we repeated this two-step process, but used the inverse of the homeless risk score for all those with risk scores less than 0.5 in the first step.

For each of the above analytic approaches, we compared the risk of fatal opioid overdose between the homeless and non-homeless groups using rate ratios with 95% confidence intervals estimated using standard techniques [28].

## Results

### Observed homelessness

Of 5,050,639 individuals in the analytic cohort, 41,457 (0.82%) were identified as experiencing homelessness according to our pre-specified indicators of known cases of homelessness. Based on ICD codes the number of individuals identified as homeless in each of the datasets used to construct this measure were as follows: 23,239 individuals in the APCD dataset, 21,722 in the CaseMix dataset, 300 in the DMH dataset, 3,237 in the MATRIS dataset, and 6,704 were identified based on the PMP. A total of 13,745 individuals, roughly one third of all those identified as homeless, were identified based on multiple indicators (See S1 Fig).

### Homelessness classification

Applying the parameters of the classification model estimated on the downsampled development sample to the validation sample yielded an AUC of 0.94, which is in the excellent range

**Table 2. Summary of model performance.**

| Metric | Value |
|---|---|
| Area under the receiver operating curve | 0.94 |
| Balanced accuracy | 86.4 |
| Sensitivity | 77.8 |
| Specificity | 95.1 |
| Positive predictive value | 11.7 |
| Negative predictive value | 99.8 |

by conventional guidelines [29]. S1 Table provides parameter estimates for the full development sample model. In assigning all individuals in the validation sample with a predicted probability of homelessness greater than or equal to 0.5, the model identified a total of 69,675 individuals in the validation sample (5.5%) as homeless. Table 2 summarizes the metrics used to assess model performance. Balanced accuracy in the validation sample was 86.4%, indicating that, on average, 86.4% of cases in each outcome category were correctly classified. Sensitivity and specificity were 77.8% and 95.1%, respectively. Positive predictive value in the validation sample was 11.7% and negative predictive value was 99.8%. Dividing this positive predictive value (or equivalently the proportion of model-predicted homeless cases that are truly homeless) by the baseline prevalence of homelessness of 0.82% (or equivalently the expected proportion of cases that would be truly homeless if they were randomly selected), indicates that the performance of the model in identifying persons experiencing homelessness was more than fourteen times better than what would be expected based on chance alone. Nonetheless, taking the reciprocal of this positive predictive value also indicates that, for each person that the model correctly identified as experiencing homelessness, there would be 8.5 false positives.

## Fatal opioid overdoses

A total of 1,265 individuals in the validation sample experienced a fatal opioid overdose during the study period, resulting in a crude opioid-related mortality rate of 100.2 per 100,000 individuals. Table 3 summarizes the results of the comparison of fatal opioid overdose rates by homeless status in the validation sample. Using model-predicted probabilities to assign individuals to a homeless status (i.e. using a predicted probability of 0.5 as the threshold), resulted in a 22.9-fold increased risk of fatal opioid overdose in the homeless group relative to the non-homeless group. The two alternative approaches yielded estimated fatal overdose rates that were, respectively, roughly 9 and 21 times higher in the homeless group than in the non-homeless group.

## Discussion

This study provides a novel approach for identifying homelessness probability using integrated administrative data from a large number of service systems. Leveraging data from these

**Table 3. Summary of fatal overdoses based on validation sample.**

| | Homeless | | | Not Homeless | | | Rate ratio (95% CI) |
|---|---|---|---|---|---|---|---|
| | N | No. of deaths | Mortality rate | N | No. of deaths | Mortality rate | |
| Model predicted homeless status | 69,675 | 724 | 1039.1 | 1,192,443 | 541 | 45.4 | 22.9 (20.5–25.6) |
| Weighted homeless status (Approach 1) | 169,378 | 743 | 438.7 | 1,093,281 | 522 | 47.7 | 9.2 (8.2–10.3) |
| Weighted homeless status (Approach 2) | 55,430 | 618 | 1114.9 | 1,207,229 | 647 | 53.6 | 20.8 (18.6–23.2) |

systems, we developed an accurate classification model with high specificity (95.4%), moderate sensitivity (77.8%), and excellent classification properties (AUC 0.94; balanced accuracy 86.4%). The strong relationship between model-predicted homeless status and several conditions with established associations with homelessness provided additional support for the validity of our model.

The strong overall performance of our model serves as a valuable proof of concept for other service systems or localities that are interested in identifying clients experiencing housing instability or homelessness even when such information is not directly available. Since we based our model on predictors obtained from a unique integrated administrative dataset, it may be difficult for other localities to replicate our exact model. Nonetheless, our results underscore the potential value of the general approach of linking data from multiple service systems and applying classification modeling techniques to these data. Doing so can lead to improved identification of housing instability and other risk factors that negatively impact health and well-being, which are difficult to measure and typically poorly captured in many service systems.

Using data in this manner carries with it the potential for the more efficient targeting of specialized service interventions at the point of care, particularly in the medical care, behavioral health, and criminal justice service systems. Our study presents a proof of concept of this idea, rather than a shelf-ready approach that can be applied immediately. However, the potential for developing such an applied approach based on our findings is real. To illustrate one example of the potential value of applying our model, we assessed the association between model-predicted homeless status and fatal opioid overdoses. We found a substantially elevated risk of fatal opioid overdose among those identified by the model as having a high probability of homelessness in alignment with prior research [19, 30]. This finding underscores the sizeable opportunity that could exist for reducing fatal overdoses if such individuals could be proactively identified and targeted for effective treatment interventions. There are analogous approaches already in use in other contexts. For example, Allegheny County in Pennsylvania uses linked administrative data from multiple county agencies in a predictive model that serves as a decision aid to frontline workers who screen and triage cases referred to the local child welfare system [31]. The model assigns risk scores quantifying both the likelihood of re-referral to the child welfare system were a worker to screen a child out of the system, and of foster care placement, were a child to be screened-in for further investigation. From a practical standpoint, moving from the development of a predictive model to the application of model results to inform service delivery requires resolving a host of technical, legal and ethical issues. Resolving these issues is not a small challenge, but neither is it an insurmountable one.

The high specificity of the model relative to its lower sensitivity indicates better performance at correctly identifying persons not experiencing homelessness than in correctly identifying those experiencing homelessness. The relatively low positive predictive value (11.7%) of our model was tied to the low prevalence of homelessness (0.82%), based on the indicators we used to identify known cases of homelessness, in the available data and underscores the challenges associated with developing predictive models for a relatively rare phenomenon. Indeed, while our model performed much better at correctly identifying persons experiencing homelessness than would be expected by chance, it nonetheless identified nearly nine false positives for every person correctly identified as homeless. As implied above, this approach may not be suitable for identifying individuals in near real-time but more useful for evaluating policies and programs aimed at serving a poorly identified population.

Finally, it is also important to note that this project was part of a larger effort to better understand predictors of fatal and nonfatal opioid overdose in Massachusetts. As such, the Chapter 55 data set was developed for multiple uses and users. The likelihood estimates of

homelessness were made available to multiple research groups for inclusion in their models or for testing their hypotheses. This model of shared knowledge has profound implications for research uses of state collected administrative data sets.

This study has several limitations. First, the indicators we used to identify known cases of homelessness likely failed to capture many individuals who actually did experience homelessness over the five-year study period. Indeed, the prevalence of homelessness observed in the present study (0.82%) is far lower than prior estimates of the five-year prevalence of homelessness in the general population (4.6%) [32]. This underestimate was due in part to the fact that identification of homelessness was conditional on use of a service system that captured information about housing status. Access to Homelessness Management Information System (HMIS) records collected on a routine basis by the homeless assistance system, would have improved the quality of our measure of homelessness although it still would have been imperfect and incomplete. The shortcomings of our measure of homelessness likely affected the performance of our model, although, at the same time, the lack of reliable measures of homelessness in the Chapter 55 data was one of the primary motivations for this study. Another limitation is that our cross-sectional approach could not take into account the duration of homelessness or its timing relative to other service use experiences (e.g. episodes of incarceration) used as predictor variables in our model. This means that some experiences used as predictors in our model may have temporally succeeded an individual's experience of homelessness. Similarly, our analysis of the relationship between model-predicted homeless status and fatal opioid overdoses was potentially biased by the inclusion of substance-use related measures in our predictive model of homelessness. Additionally, selected demographic variables known to be associated with homelessness (e.g. gender identity and sexual orientation) were not reliably available in the Chapter 55 data and were not included in our model.

## Conclusions

The present study is a useful example of how large, integrated administrative data from multiple service systems can be used to identify individuals at risk of homelessness to facilitate targeted services or timely intervention. Prior research has shown that homeless individuals have a high burden of medical and mental illnesses, substance use disorders, and health care and human services systems use [1, 33–36]. By identifying individuals at risk of homelessness, service providers can improve the coordination of services and promote better health outcomes, particularly for conditions such as opioid use disorder that exact a high toll on individuals experiencing homelessness. Future work should focus on refining and our approach to aid in identifying individuals at high risk of homelessness who may benefit from targeted service interventions.

## Supporting information

**S1 Fig. Flow diagram of sample selection.**
(DOCX)

**S1 Table. Full results of binary logistic regression model fit on development sample.**
(DOCX)

## Author Contributions

**Conceptualization:** Thomas Byrne, Travis Baggett, Thomas Land, Dana Bernson, Cheryl Kennedy-Perez, Rodrigo Monterrey, David Smelson, Monica Bharel.

**Data curation:** Thomas Byrne, Travis Baggett, Thomas Land, Dana Bernson, Monica Bharel.

**Formal analysis:** Thomas Byrne, Maria-Elena Hood.

**Investigation:** Thomas Byrne.

**Methodology:** Thomas Byrne, Travis Baggett, Thomas Land, Dana Bernson, Cheryl Kennedy-Perez, Rodrigo Monterrey, David Smelson, Marc Dones, Monica Bharel.

**Project administration:** Thomas Byrne.

**Resources:** Monica Bharel.

**Supervision:** Monica Bharel.

**Writing – original draft:** Thomas Byrne, Travis Baggett, Thomas Land, Dana Bernson, Maria-Elena Hood.

**Writing – review & editing:** Thomas Byrne, Travis Baggett, Thomas Land, Dana Bernson, Maria-Elena Hood, Cheryl Kennedy-Perez, Rodrigo Monterrey, David Smelson, Marc Dones, Monica Bharel.

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
