## [Decision Letter · Decision Letter 0]

23 Jan 2020

PONE-D-19-26052

Predicting homelessness using integrated administrative data: Implications for targeting interventions to improve the housing status, health and well-being of a highly vulnerable population

PLOS ONE

Dear Dr. Byrne,

Thank you for submitting your manuscript to PLOS ONE. After careful consideration, we feel that it has merit but does not fully meet PLOS ONE’s publication criteria as it currently stands. Therefore, we invite you to submit a revised version of the manuscript that addresses the points raised during the review process.

We would appreciate receiving your revised manuscript by Mar 08 2020 11:59PM. To enhance the reproducibility of your results, we recommend that if applicable you deposit your laboratory protocols in protocols.io, where a protocol can be assigned its own identifier (DOI) such that it can be cited independently in the future. For instructions see: http://journals.plos.org/plosone/s/submission-guidelines#loc-laboratory-protocols

We look forward to receiving your revised manuscript.

Kind regards,

Benn Sartorius, PhD

Academic Editor

PLOS ONE

Additional Editor Comments (if provided):

Editorial comments

Please include a completed GATHER checklist as part of the supplementary material and make reference to this in the methods.

Journal Requirements:

2. In your data availability statement, please add the date in the following statement: "The deadline for responding to the Notice of Opportunity was [XX].

"I have read the journal's policy and the authors of this manuscript have the following competing interests: Travis Baggett receives royalties from UpToDate for authorship of a topic review on health care for homeless people. No other authors have any competing interests to disclose." 

"I have read the journal's policy and the authors of this manuscript have the following competing interests: Travis Baggett receives royalties from UpToDate for authorship of a topic review on health care for homeless people. No other authors have any competing interests to disclose."

We note that one or more of the authors are employed by a commercial company: Future Laboratories.

Reviewers' comments:

Reviewer's Responses to Questions

**Comments to the Author**

1. Is the manuscript technically sound, and do the data support the conclusions?

Reviewer #1: Yes

Reviewer #2: Partly

2. Has the statistical analysis been performed appropriately and rigorously? 

Reviewer #1: Yes

Reviewer #2: I Don't Know

3. Have the authors made all data underlying the findings in their manuscript fully available?

Reviewer #1: No

Reviewer #2: No

4. Is the manuscript presented in an intelligible fashion and written in standard English?

Reviewer #1: Yes

Reviewer #2: Yes

5. Review Comments to the Author

Reviewer #1: This is a very interesting use of a novel linked administrative dataset at a state level. The study methods are appropriate to the data, but one question is why the authors did not use a machine learning approach (e.g. PRM)? The primary limitation of the study is that homelessness is significantly underidentified in the data available and biased to specific service users. This undoubtedly contributes to the modest sensitivity. As the authors note in the study limitations section, the study would have been greatly strengthened by inclusion of general shelter user data.

One of the indicators for homelessness was Emergency Assistance receipt, which I believe is the emergency shelter assistance program for families. If so, this means that nearly all of the homeless family adults were identified for the study, in contrast to the single adults. And given that homeless families are quite distinct from single adults, in terms of their risk factors and service use patterns, the study model may have been muddled by the combining of adults in homeless families and other homeless adults. Perhaps the model should be run separately for families and single adults to see if it improves model performance. However, based on their description, the dat may not be available to rerun the analysis in this way.

Otherwise, I found the paper to be a strong contribution, particularly in light of the fact that the model could be used by other states to identify highly vulnerable people in various service systems who could be screened for homelessness risk, and provided prevention services.

Reviewer #2: Date: 1/22/2020

Manuscript #: PONE-D-19-26052

Title: Predicting homelessness using integrated administrative data: Implications for

targeting interventions to improve the housing status, health and well-being of a highly

vulnerable population

Overall comment: Identifying (classifying or predicting) homelessness is an important topic and has many applications in healthcare and elsewhere. The paper is well-written. I have two major concerns: (1) right type of analysis and (2) focus of the paper. First, in its current format, I think the analysis represents a cross-sectional classification/association type analysis rather than a prediction study. There is no indication whether the variables used occurred before, during, or after experiencing homelessness. As a matter of fact, authors do not have a good indication on when homelessness has occurred. For a prediction or prognosis study, predictors should occur before the event. The timing is important and key in a predictive model. This is not the case in this study. Having said that, I still see value in the study. Classification or identification of risks associated with homelessness is also important. It is up to authors to decide what they want to do and appropriately conduct the analysis. Second, the paper is not focused. I would remove the association between homelessness and other health condition and opioid overdose from the paper. They are irrelevant to the main topic. They can be presented separately and more in depth elsewhere.

The followings are my minor comments:

Abstract:

The main purpose of the study was to develop and validate a model for homelessness prediction. This is an important topic. However, I am not sure why the authors lost the focus and brought into attention association between homelessness and a series of health conditions including opioid overdose. I would suggest the authors discuss these as potential applications of their predictive model and not as the main focus of the manuscript.

Introduction:

Page 3, line 56: Change “service systems” to “publicly-funded systems.”

Page 4, lines 83-86: what is the basis for your assumption? Any citation to validate or explain how you made this assumption and how accurate it could be?

Page 4, lines 86-88: Again, what is the basis for this claim? This is a huge assumption to make. You are basically validating your model not based on actual data on homelessness but based on measures that correlates with homelessness. What are the degrees of correlations? Elaborate. Cite your references for such assumption.

Page 4. Lines 88-91: As I mentioned above, I strongly recommend keeping the paper focus. This paper is about a predictive model of homelessness based on integrated administrative data. Keep the rest for future papers. And stay focus on using various predictive models, make your model parsimonious, validate it properly, show its economic usefulness, etc., etc.

Data and Sample:

Page 5, line 97: What are programmatic decisions?

Page 5, lines 104-105: What are the 15 data sets? What variables is linked with the main Chapter 55 dataset? Why these variables are chosen? Cite your multistage deterministic approach to merge the data across all these datasets (or put it in the appendix).

Page 5, line 114: consider rewording the sentence. So, did I understand this correctly? Among 14,245,349 people included in the Chapter 55 dataset only 5,050,639 had a record in the APCD. Please include a complete and detailed schematic flow diagram of your sample size. This can be included in the appendix.

Measure of Homelessness:

Page 6, line 122-129. Please include number of homeless people identified based on each of the defined criterion in your schematic flow diagram.

Analysis:

Page 10, lines 155: Explain your stratified random sampling. How did you stratify? Did you consider other variables such as age, sex, race/ethnicity to be randomly distributed in both development and validation group?

Page 14, line 233: How did you calculate 14 times? For each correctly identified homeless person, there are 8.5 false positive. Elaborate on this.

Note: I would like to see a table with all related diagnostic measures (i.e. C-statistic, sensitivity, specificity, PPV, NPV, etc.)

Note 2. The specificity of your model is extremely high (95.1%). Could it be because of timing of prediction, meaning that you included variables in your prediction model that was taken after a person experienced homelessness. So, your model actually did not predict homelessness. It assesses the risks of several variables and their associations with homelessness. This is different from prediction. Timing is important in a predictive model. Timing of prediction should be prior to the event. So, in building a predictive model, one should use only predictors that are available prior to being homeless.

Note 3. Please include the following information for the logistic regression model in the appendix:

1. Full name of the abbreviated variables.

2. Diagnostic testing of your regression model.

3. Instead of using “unknown” or “missing” as your reference category, please use more meaningful groups for your categories. For example, for race, use “White” as your reference category.

4. This study covers a wide age-range group (11+). The question about one’s mother’s occupation for certain age group seems irrelevant. And, this variable may change frequently for certain jobs.

5. These data are gathered over time. What if the condition for one person changed? Any thought of including longitudinal (time-variant) variables in your model? In that case probably a generalized estimating equation would be more appropriate than simple logistic model.

6. What do these varibles mean or represent?

Any record in BSAS

Any record in Casemix mental health records

Any record in DMH

Any record in DVS

Any record in Matris

Any record in PMP

a.

6. PLOS authors have the option to publish the peer review history of their article (what does this mean?). If published, this will include your full peer review and any attached files.

Reviewer #1: No

Reviewer #2: No

---

## [Author Response · Author response to Decision Letter 0]

15 Jul 2020

We thank the reviewers for their thorough feedback on this manuscript. We have taken their feedback into account in revising this manuscript, and believe it is stronger as a result. Below, we detail how we responded to each of the comments offered by the reviewers.

Reviewer #1 

1. This is a very interesting use of a novel linked administrative dataset at a state level. The study methods are appropriate to the data, but one question is why the authors did not use a machine learning approach (e.g. PRM)? The primary limitation of the study is that homelessness is significantly underidentified in the data available and biased to specific service users. This undoubtedly contributes to the modest sensitivity. As the authors note in the study limitations section, the study would have been greatly strengthened by inclusion of general shelter user data.

We thank the reviewer for this positive assessment of our use of this unique dataset. We appreciate the reviewer’s suggestion that we use a machine learning approach as part of our analytic strategy. We did, in fact hope to employ machine learning algorithm approaches such as random forests or support vector machines as an alternative and point of comparison to logistic regression. Unfortunately, due to the unique nature of these data, they could only be managed and analyzed using a version of the SAS statistical software that did not provide us with the capability to use such algorithms. However, it bears mentioning that recent work by Gao and colleagues (2017) and the present study’s lead author (Byrne et al., 2019) have shown that the performance of machine learning algorithms in predicting homelessness is only marginally better than logistic regression. 

We also agree that the study could be strengthened by the inclusion of general shelter use data. Unfortunately, these data were unavailable in part because data for the single adult shelter system are collected and maintained not by a single state agency, but by more than a dozen distinct entities—known as Continuums of Care (CoCs) in various regions throughout the state. Therefore was not possible to obtain data from these entities and merge them with the chapter 55 database. 

Gao, Y., Das, S., & Fowler, P. (2017, March). Homelessness service provision: a data science perspective. In Workshops at the Thirty-First AAAI Conference on Artificial Intelligence.

Byrne, T., Montgomery, A. E., & Fargo, J. D. (2019). Predictive modeling of housing instability and homelessness in the Veterans Health Administration. Health services research, 54(1), 75-85.

2. One of the indicators for homelessness was Emergency Assistance receipt, which I believe is the emergency shelter assistance program for families. If so, this means that nearly all of the homeless family adults were identified for the study, in contrast to the single adults. And given that homeless families are quite distinct from single adults, in terms of their risk factors and service use patterns, the study model may have been muddled by the combining of adults in homeless families and other homeless adults. Perhaps the model should be run separately for families and single adults to see if it improves model performance. However, based on their description, the dat may not be available to rerun the analysis in this way.

The reviewer raises an excellent point with this comment. It is indeed correct that records from the Emergency Assistance program was one of the data sources included in the Chapter 55 data warehouse upon which the study was based. However, we did not use receipt of Emergency Assistance services an indicator of homelessness for precisely the reason that the reviewer mentions: it would have led to a nearly completely accurate identification of homelessness among adults in families, and a far less accurate identification of homelessness among single adults. We used the Emergency Assistance data only as way to restrict the sample to eliminate what were assumed to be duplicate records in the All Payer Claims Database, which served as the base dataset for the data linkage, i.e. we only included persons who had a claim in the All Payer Claims Database AND at least one of the other data sources (one of which was Emergency Assistance data). As we note in the methods section we used the following indicators to identify homelessness: 1) a claim in the APCD or record in the Acute Care Hospital Case Mix (Case Mix) data with an accompanying ICD-9 V.60 or ICD-10 Z590 code indicating homelessness; 2) a record in the Department of Mental Health (DMH) dataset in which individuals were ever identified as experiencing a loss of housing based on a measure of housing status captured on a monthly basis for all DMH clients; 3) an ambulance record in the Massachusetts Ambulance Trip Record Information System (MATRIS) data in which the word “homeless” or “shelter” appeared in the narrative report; or 4) a prescription record in the Prescription Monitoring Program (PMP) in which the patient’s address matched that of an emergency shelter. 

3. Otherwise, I found the paper to be a strong contribution, particularly in light of the fact that the model could be used by other states to identify highly vulnerable people in various service systems who could be screened for homelessness risk, and provided prevention services.

We thank the reviewer for this generous assessment of our work. While we do not view our analysis as delivering a “shelf ready” product that can be used to identify those at risk of homelessness and target them with prevention services accordingly, we do believe that it offers “proof of concept” of how integrated administrative data might be used for this purpose. 

Reviewer #2 

1. Overall comment: Identifying (classifying or predicting) homelessness is an important topic and has many applications in healthcare and elsewhere. The paper is well-written. I have two major concerns: (1) right type of analysis and (2) focus of the paper. First, in its current format, I think the analysis represents a cross-sectional classification/association type analysis rather than a prediction study. There is no indication whether the variables used occurred before, during, or after experiencing homelessness. As a matter of fact, authors do not have a good indication on when homelessness has occurred. For a prediction or prognosis study, predictors should occur before the event. The timing is important and key in a predictive model. This is not the case in this study. Having said that, I still see value in the study. Classification or identification of risks associated with homelessness is also important. It is up to authors to decide what they want to do and appropriately conduct the analysis. 

We thank the reviewer for this comment. While the terms “predictive model” are “classification model” are frequently used interchangeably, we concede the reviewer’s point that our use of the term “predictive model” may be misleading. Additionally, while we would like to revise our analysis to take temporal ordering into account, we cannot accurately assess onset of homelessness in our data, but rather can only assess when someone is identified as homeless in one of the administrative data sources we use. The distinction is subtle, but important in terms of developing a truly “predictive” model. In other words, we could theoretically estimate a model in which we use predictors collected only prior to a first indicated date of homelessness, but this too would be problematic because we could not actually assess whether predictors temporally preceded the onset of an episode of homelessness. As such, we have decided to retain our cross-sectional approach, and describe this is a limitation in the Discussion. We also now use the term “classification model” throughout the paper and explain our reasoning for doing so in the Analysis section where we have added the following explanation:

Our analytic plan centered on the development and testing of a classification model of homelessness. The terms “predictive model” and “classification model” are frequently used interchangeably to describe models that attempt to predict a categorical outcome conditional on a set of predictors with the goal of maximizing the performance of such models. Although these terms are often used interchangeably, we chose to use the term "classification model" in the current paper to reflect the fact that our data are cross-sectional, and thus we cannot determine the temporal ordering of our outcome variable (homelessness) relative to our set of independent variables. Thus, our analysis is not “predicting” a future outcome on a set of antecedent predictors, and we use the term “classification” model to avoid confusion about the scope of our analysis. 

2. Second, the paper is not focused. I would remove the association between homelessness and other health condition and opioid overdose from the paper. They are irrelevant to the main topic. They can be presented separately and more in depth elsewhere.

We thank the reviewer for this comment and recognize that the inclusion of these other health outcomes made for a somewhat disjointed paper in its original configuration. As such, we have removed the analysis of the association between homelessness and the health conditions (e.g. Hepatitis C, HIV/AIDS) that we had originally included as validators of our model. We have, however, retained the analysis of the association between model-predicted homeless status and fatal opioid overdoses, and reframed the presentation of this analysis. Our decision to retain this analysis was motivated by two factors. First, the regulations put in place by the Massachusetts legislature governing the use of the Chapter 55 data used in this study required that all analyses using the data include some examination of opioid overdoses as part of their scope. Second, and more important, we believe that there is substantive value in retaining this analysis in the paper even if these regulations did not exist. Specifically, as we try to emphasize in the Discussion, we envision our analysis not simply as an academic exercise, but as the first step towards an applied use of more refined modeling approaches that would lead to targeting of services to improve housing and health outcomes among a high-risk population. Thus, our inclusion of an analysis of the relationship between model-predicted homeless status and fatal opioid overdoses is intended as an example that demonstrates the potential value and opportunity that exists in targeting interventions in this manner to mitigate the risk of a serious adverse health outcome. To more clearly articulate this rationale for including this analysis, we have reframed our description of the aim assessing the relationship between model-predicted homeless status and fatal opioid overdoses in the Introduction and the implications of the results of this analysis in the Discussion. 

The followings are my minor comments:

3. Abstract:

The main purpose of the study was to develop and validate a model for homelessness prediction. This is an important topic. However, I am not sure why the authors lost the focus and brought into attention association between homelessness and a series of health conditions including opioid overdose. I would suggest the authors discuss these as potential applications of their predictive model and not as the main focus of the manuscript.

We thank the reviewer for this comment. As we noted above, we have revised the paper to 1) remove the analysis of the relationship between model-predicted homeless status and health conditions we had previously included as validators; and 2) reframed our inclusion of the analysis of the relationship between model predicted homeless status and fatal opioid overdoses. We have amended the abstract to reflect these changes. 

4. Introduction:

Page 3, line 56: Change “service systems” to “publicly-funded systems.”

We have made this change.

5. Page 4, lines 83-86: what is the basis for your assumption? Any citation to validate or explain how you made this assumption and how accurate it could be?

We thank the reviewer for this comment. We have revised this sentence to explain the basis of this assumption. Specifically, the assumption is rooted in long-standing challenges in accurately identifying people experiencing homelessness using administrative data and the fact that even administrative data with information on housing status may not accurately capture an individual’s true housing status. We now explain these points and have included citations to justify then. 

6. Page 4, lines 86-88: Again, what is the basis for this claim? This is a huge assumption to make. You are basically validating your model not based on actual data on homelessness but based on measures that correlates with homelessness. What are the degrees of correlations? Elaborate. Cite your references for such assumption.

We agree with this comment and, as described above, we have removed the analysis in which we seek to validate our models against other health conditions. 

7 .Page 4. Lines 88-91: As I mentioned above, I strongly recommend keeping the paper focus. This paper is about a predictive model of homelessness based on integrated administrative data. Keep the rest for future papers. And stay focus on using various predictive models, make your model parsimonious, validate it properly, show its economic usefulness, etc., etc.

We thank the reviewer again for this comment. As noted above, we have made changes to the paper to more clearly focus on the classification model and to make a more explicit connection between the model and our analysis of the relationship between model-predicted homeless status and fatal opioid overdoses. 

8. Data and Sample:

Page 5, line 97: What are programmatic decisions?

We have removed this phrase and now leave mention the goal to “guide policy development” which we feel is more clear. 

9. Page 5, lines 104-105: What are the 15 data sets? What variables is linked with the main Chapter 55 dataset? Why these variables are chosen? Cite your multistage deterministic approach to merge the data across all these datasets (or put it in the appendix).

Thank you for raising this comment. The 15 data sets we used are all included in Table 1 and we now direct readers to that Table in this section of the paper. We believe a full discussion of the multistage deterministic matching approach is beyond the scope of the current paper, although it has been described in full detail in a publicly available report from the Massachusetts Department of Public Health. We thus cite this report and direct interested readers to it by writing the following (new text shown in italics): 

APCD data for these individuals, who represent more than 98% of Massachusetts residents, were linked with other datasets using a multistage deterministic linkage algorithm; full details on this linkage algorithm and about the 16 data sources included in the Chapter 55 data warehouse are available elsewhere.[19] 

10. Page 5, line 114: consider rewording the sentence. So, did I understand this correctly? Among 14,245,349 people included in the Chapter 55 dataset only 5,050,639 had a record in the APCD. Please include a complete and detailed schematic flow diagram of your sample size. This can be included in the appendix.

We have revised the wording to clarify that there are actually 14,245,349 unique records in the APCD, but we only included the 5,050,639 who also had at least one record in a dataset besides the APCD. The remaining records are suspected duplicates/out of state residents. We have also now included the recommended schematic diagram as S1 Figure.

11. Measure of Homelessness:

Page 6, line 122-129. Please include number of homeless people identified based on each of the defined criterion in your schematic flow diagram.

We have included this in the newly added S1 Figure

12. Analysis:

Page 10, lines 155: Explain your stratified random sampling. How did you stratify? Did you consider other variables such as age, sex, race/ethnicity to be randomly distributed in both development and validation group?

We have revised this section to more clearly describe the stratified random sampling procedure we used and why we used it. Specifically, we now explain that we used our measure of whether an individual was identified as homeless to stratify our sample into two strata. We then randomly sampled 75% of cases within each stratum to be assigned to the development sample and the remaining 25% within each stratum were assigned to the validation sample. Given the small number of cases in our overall sample identified as homeless and attendant risk that simple randomly sampling would result in a large proportion of homeless cases in the development (or validation) sample, we used this approach to balance the proportion of homeless cases across the development and validation sample. 

13. Page 14, line 233: How did you calculate 14 times? For each correctly identified homeless person, there are 8.5 false positive. Elaborate on this.

We now explain how we calculated each of these. 

14. Note: I would like to see a table with all related diagnostic measures (i.e. C-statistic, sensitivity, specificity, PPV, NPV, etc.)

We now include these in a new table, Table 2. 

15. Note 2. The specificity of your model is extremely high (95.1%). Could it be because of timing of prediction, meaning that you included variables in your prediction model that was taken after a person experienced homelessness. So, your model actually did not predict homelessness. It assesses the risks of several variables and their associations with homelessness. This is different from prediction. Timing is important in a predictive model. Timing of prediction should be prior to the event. So, in building a predictive model, one should use only predictors that are available prior to being homeless.

We acknowledge that this is a limitation of our paper. As we note above, and in response to similar feedback offered by the reviewer, we have now amended the description of our modeling approach to reflect the fact that it is not possible to assess temporal ordering between our predictors and our outcomes.

16. Note 3. Please include the following information for the logistic regression model in the appendix:

 1. Full name of the abbreviated variables.

 We have now added a note in which we spell out all abbreviations used in the table. 

 2. Diagnostic testing of your regression model.

We are not sure what specific diagnostics the reviewer is hoping to see. We are happy to include them if the reviewer can be more specific. 

3. Instead of using “unknown” or “missing” as your reference category, please use more meaningful groups for your categories. For example, for race, use “White” as your reference category.

Due to restrictions around the use of the Chapter 55 data, we were unable to re-estimate the models changing the reference categories. However, while potentially of substantive interest, the reference category should not affect the performance of the models, which was the main focus of the analysis, and not the substantive relationships. 

4. This study covers a wide age-range group (11+). The question about one’s mother’s occupation for certain age group seems irrelevant. And, this variable may change frequently for certain jobs.

We agree that this is potentially a limitation. However, given that the goal of the analysis was to maximize model performance, we erred on the side of being overly inclusive with predictors. 

5. These data are gathered over time. What if the condition for one person changed? Any thought of including longitudinal (time-variant) variables in your model? In that case probably a generalized estimating equation would be more appropriate than simple logistic model.

In the ideal case, we would include these. Unfortunately, as we outline above, given that our data do not allow for a true assessment of the timing of the onset of homelessness, we believe our more conservative cross-sectional approach is preferred. We note the limitations associated with our cross-sectional approach in the Discussion. 

6. What do these varibles mean or represent?

Any record in BSAS

Any record in Casemix mental health records

Any record in DMH

Any record in DVS

Any record in Matris

Any record in PMP

We have amended the description of these variables in the table to clarify that the refer to whether an individual had received any service from each of the agencies.

---

## [Editor Report · Decision Letter 1]

6 Aug 2020

A classification model of homelessness using integrated administrative data: Implications for targeting interventions to improve the housing status, health and well-being of a highly vulnerable population

PONE-D-19-26052R1

Dear Dr. Byrne,

We’re pleased to inform you that your manuscript has been judged scientifically suitable for publication and will be formally accepted for publication once it meets all outstanding technical requirements.

Kind regards,

Benn Sartorius, PhD

Academic Editor

PLOS ONE
---

## [Editor Report · Acceptance letter]

11 Aug 2020

PONE-D-19-26052R1 

A classification model of homelessness using integrated administrative data: Implications for targeting interventions to improve the housing status, health and well-being of a highly vulnerable population 

Dear Dr. Byrne:

I'm pleased to inform you that your manuscript has been deemed suitable for publication in PLOS ONE. Congratulations! Your manuscript is now with our production department. 

Kind regards, 

on behalf of

Dr. Benn Sartorius 

Academic Editor

PLOS ONE